# Use of Essential Oil Emulsions to Control *Escherichia coli* O157:H7 in the Postharvest Washing of Lettuce

**DOI:** 10.3390/foods12132571

**Published:** 2023-06-30

**Authors:** Jessica Santos Pizzo, Raira Andrade Pelvine, Andre Luiz Biscaia Ribeiro da Silva, Jane Martha Graton Mikcha, Jesui Vergilio Visentainer, Camila Rodrigues

**Affiliations:** 1Department of Horticulture, Auburn University, Auburn, AL 36849, USA; jzd0148@auburn.edu (J.S.P.); rza0071@auburn.edu (R.A.P.); adasilva@auburn.edu (A.L.B.R.d.S.); 2Center for Agricultural Sciences, Postgraduate Program of Food Science, State University of Maringa, Maringa 87020-900, PR, Brazil; jmgmikha@uem.br; 3Department of Chemistry, State University of Maringa, Maringa 87020-900, PR, Brazil; jesuiv@gmail.com

**Keywords:** produce safety, sanitizer, chlorine, peroxyacetic acid, lettuce, postharvest wash, emulsified essential oils, cross-contamination

## Abstract

Essential oils (EOs) have strong antibacterial properties and can be potential sanitizers to reduce pathogen load and prevent cross-contamination during postharvest washing. The objective of this study was to investigate the efficacy of emulsions containing oregano (OR; *Origanum vulgare*) and winter savory (WS; *Satureja montana*) EOs at different concentrations (0.94 and 1.88 µL/mL) and storage times (0 h, 24 h, and 7 days), in reducing *Escherichia coli* O157:H7 on the surface of three types of lettuce (romaine, crisphead, and butterhead). The EO emulsions were compared with one no-rinse treatment and three rinse treatments using water, 200 ppm chlorine, and 80 ppm peroxyacetic acid (PAA), respectively, in a simulated washing system. The results showed that while the EO emulsions significantly reduced *E. coli* O157:H7 on crisphead lettuce over time, not all treatments were effective for romaine and butterhead lettuce. The mixture of OR and WS at concentrations of 0.94 and 1.88 µL/mL was found to be the most effective in reducing *E. coli* O157:H7 on inoculated lettuce, resulting in reductions of 3.52 and 3.41 log CFU/g, respectively. Furthermore, the PAA and the mixture of OR and WS at 1.88 µL/mL effectively limited bacterial cross-contamination close to the detection limit for all lettuce types during all storage times. These results suggest that OR and WS EOs could serve as potential alternatives to chemical sanitizers for postharvest lettuce washing.

## 1. Introduction

Lettuce (*Lactuca sativa*) is a vegetable crop grown across the entire United States, which is ranked as the second largest lettuce producer in the world. However, foodborne outbreaks associated with lettuce contamination by *Escherichia coli* O157:H7 have increased over the past few years in the United States, leading to serious health risks for consumers and highlighting the need for effective control measures to prevent or reduce lettuce contamination by these pathogens [1]. Lettuce contamination by foodborne pathogens, including *E. coli* O157:H7, occurs from various sources, including animals, biological soil amendments of animal origin, water, soil, humans, and equipment. Contamination may occur at several points during the production chain, including harvest, postharvest, storage, and transportation [2]. Consequently, using good agricultural practices throughout the food production chain is critical for preventing and controlling the microbial contamination of fresh produce [3]. The use of chemical sanitizers during postharvest produce washing is the most common strategy used to minimize cross-contamination in lettuce production systems [3]. 

Chlorine and peroxyacetic acid (PAA) are the most commonly used sanitizers for postharvest washing due to their known antimicrobial activity and ease of application. However, if mishandled, chlorine and PAA can pose a significant health risk to workers [4]. In addition, they are not very effective in controlling pathogens attached to produce surfaces, often resulting in reductions of only 1.0–2.5 log CFU/g [5,6,7,8]. Thus, the development of an effective decontamination method is required to reduce *E. coli* O157:H7 contamination on the surface of the fresh produce. In this context, natural antimicrobials have emerged as a potential alternative to traditional sanitizers, such as chlorine and PAA, for postharvest washing of fresh produce, including lettuce.

Natural antimicrobials, particularly essential oils (EOs), are known for their antimicrobial properties and have been studied as alternatives to chemical sanitizers against foodborne pathogens, including *E. coli*, *Salmonella*, and *Listeria monocytogenes* [7,9,10,11,12,13,14,15]. The antimicrobial activity of EOs is attributed to the presence of bioactive compounds, such as terpenes and phenolics, which have been found to be effective against various microorganisms [8,10,11,12,13,14,15,16]. For example, the major component of oregano EO is carvacrol, with concentrations ranging from 61 to 85% [16,17,18,19,20], and it has been shown to be effective against various foodborne pathogens, including *E. coli* O157:H7 [21]. Similarly, winter savory EO contains carvacrol at similar ranges (55 to 89%) as the main component and also has antimicrobial activity against foodborne pathogens [18,22,23,24,25].

The mechanism of action of EOs as antimicrobials is complex and involves various modes of action. These include the disruption of the cell membrane due to damage to the lipid bilayer, which results in the leakage of intracellular constituents; inhibition of metabolic processes; and interference with enzyme activity [21,26,27,28,29]. Moreover, the synergistic effects among the bioactive compounds present in EOs are thought to contribute to their antimicrobial activity. Studies have reported that combining different EOs may exhibit a more potent inhibitory effect than individual EOs [21,26,30,31,32,33,34]. 

In general, EOs have great potential to be a safe and environmentally friendly sanitizer for lettuce due to their inclusion under the generally recognized as safe (GRAS) category by the United States Food and Drug Administration (FDA) [35]. However, the hydrophobic nature of EOs makes their direct use in washing systems difficult [36]. One strategy that could be used to improve the efficacy of EOs in washing systems is emulsified EO solutions, which stabilize EO compounds and enhance their microbial activity [14].

Therefore, the objective of this study was to evaluate the antimicrobial efficacy of emulsions of winter savory EO and oregano EO in comparison with one no-rinse treatment and three rinse treatments using water, chlorine, and PAA, respectively, in a simulated washing system against *E. coli* O157:H7 and using three lettuce types (romaine, butterhead, and crisphead). Bacterial reduction in surface-inoculated lettuce and cross-contamination after each treatment were assessed at 0 h, 24 h, and 7 days of storage.

## 2. Materials and Methods

### 2.1. Preparation of EO Emulsion

The oil-in-water emulsions were prepared according to Ghosh et al. [37] with some modifications. The winter savory EO (*Satureja montana*; Essences Bulgaria; JPV Middendorf Ltd., Ruse, Bulgaria) and oregano EO (*Origanum vulgare*; dōTERRA^®^; Pleasant Grove, UT, USA) were combined in different proportions (3:2, 2:3, 3:1, 1:3, and 1:1, respectively). Each supplier provided a gas chromatography analysis to ensure the chemical composition of each EO. The winter savory EO used in this study was mainly composed of carvacrol (64.16%), gamma-terpinene (12.35%), and p-cymene (5.27%); oregano was mainly composed of carvacrol (71.17%), gamma-terpinene (5.20%), linalool (4.84%), and p-cymene (4.02%). Each secondary compound from both oils comprised less than 2% of the total abundance. Tween 80 (Sigma-Aldrich, Saint Louis, MO, USA) was mixed in a 1:2 (*v*/*v*) mass ratio and added to deionized water. The final EO concentration in the emulsions was 30 µL/mL. Additionally, a control sample (Tween 80 + water) was prepared.

All the emulsions were prepared in two stages. First, the mixture was homogenized using a laboratory vortex mixer (S0200; Labnet International Inc., Woodbridge, NJ, USA) at a maximum speed of 10 (3400 rpm) for 2 min, with a sample volume of 50 mL each time, which led to the formation of a coarse emulsion. Then, the emulsion was sonicated for 10 min in a water bath (FS14; Fisher Scientific, Fair Lawn, NJ, USA) to form a stable emulsion.

### 2.2. Preparation of Bacteria Culture

*E. coli* O157:H7 (American Type Culture Collection, ATCC 43895) was obtained from the Department of Horticulture at Auburn University (Auburn, AL, USA). Before use, *E. coli* O157:H7 was made resistant to 50 ppm nalidixic acid (Sigma-Aldrich, Saint Louis, MO, USA) and 50 ppm rifampicin (Sigma-Aldrich, Saint Louis, MO, USA) in a stepwise procedure. Briefly, a loopful of the culture was inoculated into tryptic soy broth (TSB; BD Difco, Sparks, MD, USA) while increasing the concentration of nalidixic acid by 10 ppm with consecutive transfers every 24 h in TSB until the culture was resistant at a concentration of 50 ppm. Bacterial cultures were adapted to 50 ppm rifampicin using the same stepwise adaptation until the bacteria were resistant to both 50 ppm nalidixic acid and 50 ppm rifampicin in TSB (TSBRN). Then, a 10 µL loopful of antibiotic-adapted strain was transferred to TSBRN and incubated for 24 h at 37 °C. A 10 µL loopful was transferred to fresh the TSBRN twice before use [4].

### 2.3. Minimum Inhibitory and Minimum Bactericidal Concentration

The minimum inhibitory concentration (MIC) and minimum bactericide concentration (MBC) of EO emulsions with winter savory EO, oregano EO, and combined at different proportions (3:2, 2:3, 3:1, 1:3, and 1:1, respectively) against *E. coli* O157:H7 were determined using the broth microdilution method in 96-well microtiter plates with flat-bottom wells according to the Clinical and Laboratory Standards Institute (CLSI) guidelines [38].

The bacterial culture was adjusted to a 0.5 McFarland scale and diluted with TSB to obtain a working culture of 1.5 × 10^7^ colony-forming units (CFU)/mL. The adjusted bacterial culture (5 µL) was then inoculated into each well of a 96-well plate containing 100 µL of EO emulsion (30 µL/mL) serially diluted in 100 µL Mueller–Hinton Broth (MHB; Hardy Diagnostics, Santa Maria, CA, USA). The final concentration of bacteria was 1.5 × 10^5^ CFU/mL. The 96-well plate was incubated for 24 h at 37 °C, and the absorbance was measured at 595 nm using an iMark microplate reader (Bio-Rad, Hercules, CA, USA). The MIC was determined as the lowest concentration of EO emulsion that inhibited the growth of the bacteria with an absorbance value <0.05 at 595 nm. 

The MBC was determined by subculturing aliquots (20 µL) from wells where bacterial growth could not be observed and inoculated into Petri dishes containing tryptic soy agar (TSA; BD Difco, Sparks, MD, USA) with 50 ppm nalidixic acid and 50 ppm rifampicin (TSARN). The plates were then incubated at 37 °C for 24 h. MBC was determined as the lowest EO emulsion concentration, resulting in a ≥99.9% reduction in the initial bacterial inoculum.

MHB-only and MHB-containing bacterial cultures were used as negative and positive controls, respectively. MHB-containing blank samples (Tween 80 + water without emulsified EO) were evaluated to assess the possible antimicrobial effect of Tween 80. Each test was performed in triplicate.

### 2.4. Inoculation of Bacteria Culture on Lettuce Leaves

Fresh leaves of three different lettuce types, romaine (Monte Carlo), butterhead (Salanova^®^), and crisphead (Grazion), were obtained from the Department of Horticulture at Auburn University (Auburn, AL, USA) as part of a variety trial experiment. Lettuce leaves were washed in deionized water to remove dust and soil and then air-dried at room temperature (25 °C). Lettuce leaves were placed on a sterilized aluminum foil tray and exposed to ultraviolet light at 254 nm for 10 min on each side to reduce naturally occurring microorganisms. Lettuce leaves were stored in sterile Whirl-Pak^®^ bags (Nasco, Madison, WI, USA) at 4 °C until analysis [4]. 

The bacterial culture was poured into a sterile glass dish and adjusted to a McFarland scale of 0.5 using approximately 300 mL of phosphate-buffered solution (PBS) [39]. The final bacterial culture concentration was approximately 10^7^ CFU/mL of *E. coli* O157:H7. 

Lettuce leaves were submerged in the inoculum without agitation, using sterile tweezers to ensure complete coverage. After 2 min, the leaves were removed from the inoculum with sterile tweezers and placed in a sanitized salad spinner to remove excess liquid. The leaves remained in the biosafety cabinet and air-dried for 2 h [4]. A sample of an inoculated lettuce leaf that received no treatment (no-rinse, NR) was analyzed to determine the *E. coli* O157:H7 concentration attached to the lettuce leaves. The NR sample was prepared in triplicate.

### 2.5. Preparation of Washing Treatments

The respective washing treatments were prepared using 200 ppm of chlorine, 80 ppm PAA, and 0.94 and 1.88 µL/mL EO emulsions. The emulsion concentrations were selected based on previous MIC results. EO samples at 0.94 µL/mL were coded as “1”, whereas samples at 1.88 µL/mL were coded as “2”, resulting in the samples OR1, OR2, WS1, WS2, OR:WS1, and OR:WS2. Sterile water was used as a control to simulate washing treatment in the absence of sanitizer in the system. 

The chlorine sanitizer was prepared by diluting Clorox Regular Bleach with sterile deionized water to a final concentration of 200 ppm according to the label instructions. The final solution’s pH (Cole-Parmer pH meter—Model 05669-00) was adjusted to 7.0 (±0.02) using a 1 M citric acid solution. The PAA solution was prepared by diluting Sanidate 5.0 with sterile deionized water to a final concentration of 80 ppm according to the label instructions.

### 2.6. Simulated Postharvest Washing

The simulated postharvest washing (Figure 1) was performed according to Cimowsky et al. [4]. Briefly, surface-inoculated lettuce samples receiving the washing treatment were immersed in a sterile glass dish containing 200 mL of wash solution for 2 min without agitation. Samples were removed from the wash solution using sterile tweezers, placed into sterile Whirl-Pak^®^ bags, and diluted 1:5 (*w*/*v*) with PBS containing 0.2% Tween 80 and 0.1% sodium thiosulfate (Fisher Scientific, Fair Lawn, NJ, USA). The lettuce leaves were hand-massaged for 20 s, and the rinsate was serially diluted in buffered peptone water (BPW; Oxoid, CM 509; Fisher Scientific, Fair Lawn, NJ, USA) and spread-plated onto TSARN plates using a sterile L-spreader. The plates were incubated at 37 °C for 24 h, and the colonies were counted. Each wash treatment was performed in triplicate. The results were expressed as log CFU/g.

### 2.7. Simulated Storage

Samples for 24 h and 7 days of storage were washed as previously described. Lettuce leaves were placed in sterile Whirl-Pak^®^ bags and stored in a refrigerator at 4 °C. After storage, samples were diluted, hand-massaged, and plated as previously described in simulated postharvest treatment (Figure 1) [4]. 

### 2.8. Cross-Contamination in Non-Inoculated Lettuce Leaves

The cross-contamination analyses (Figure 1) were performed according to Cimowsky et al. [4]. For each washing treatment, after washing the inoculated lettuce leaves, one non-inoculated (NI) sample was immersed in the same washing solution to assess cross-contamination from the initial contaminated sample. The NI sample was immersed for 2 min, removed with sterile tweezers, placed in a sterile Whirl-Pak^®^ bag, diluted 1:5 (*w*/*v*) with PBS containing 0.2% Tween 80 and 0.1% sodium thiosulfate, and hand-massaged for 20 s. The rinsate was successively diluted in BPW and spread-plated onto TSARN. After incubation at 37 °C for 24 h, the colonies were counted.

### 2.9. Statistical Analysis

A generalized linear mixed method model was performed using R Studio [40]. The inoculated and NI samples were analyzed using washing treatments, lettuce types, storage duration, and their interactions as fixed effects. In both models, Tukey’s test with a significance level of 0.05 was performed when the F-value was significant. In addition, orthogonal contrasts were applied to assess the effects of washing treatments versus sterile water and NR and sterile water versus NR for inoculated samples.

## 3. Results

### 3.1. Minimum Inhibitory and Minimum Bactericidal Concentration

The oil-in-water emulsions prepared with winter savory EO, oregano EO, and their combinations in different proportions (3:2, 2:3, 3:1, 1:3, and 1:1, respectively) at a final EO concentration of 30 µL/mL were tested for their antimicrobial activity against *E. coli* O157:H7 to establish the MIC and MBC (Table 1).

The MIC value measured for WS against E. coli O157:H7 was 1.88 µL/mL, which was higher compared to all other treatments (MIC = 0.94 µL/mL), indicating that the other treatments had higher antimicrobial activity than WS. In general, the MBC values varied between 0.94 µL/mL (OR) and >30 µL/mL (WS). Tween 80 + water did not affect bacterial growth.

The MIC and MBC values of the OR:WS emulsions were lower than those of WS, suggesting an additive effect of oregano EO and winter savory EO against *E. coli* O157:H7. Based on these results, the OR:WS (2:3) treatment was selected for use in the simulated wash system due to the low MIC and MBC values compared to other mixtures of OR and WS. The concentrations of WS, OR, and OR:WS at 0.94 µL/mL and 1.88 µL/mL of EO were selected for the wash treatment to evaluate their antimicrobial efficacy against *E. coli* O157:H7 on inoculated lettuce.

### 3.2. Effects of Treatments on Inoculated Lettuce Leaves over Time

The population of *E. coli* O157:H7 and the effects of chemical treatments compared to NR and water controls on inoculated lettuce leaves after washing are shown in Table 2. The log reductions in *E. coli* O157:H7 compared to the NR control are presented in Appendix A.

There were significant three- and two-way interactions among treatments, lettuce types, and storage durations. The initial population of *E. coli* O157:H7 attached to the surface of lettuce leaves varied between 4.45 and 5.80 log CFU/g (Table 1). The maximum reductions in bacteria counts were 3.52 and 3.41 log CFU/g (compared to NR) by OR:WS1 and OR:WS2 after 24 h washing of inoculated crisphead lettuce, respectively (Appendix A).

Among the three lettuce types, the butterhead lettuce presented the highest level of *E. coli* O157:H7 recovered from surface-inoculated lettuce for all treatments and storage times, except for the 80 ppm PAA treatment after 24 h. For this treatment and time point, romaine lettuce had a higher recovered bacterial population compared to other lettuce types (Table 2).

All of the sanitizer treatments effectively reduced the initial microbial counts compared to the NR control at 0 h, 24 h, and 7 days for the inoculated romaine and crisphead lettuce. However, the effectiveness of treatments for the inoculated butterhead lettuce varied over time. At 0 h, the washing treatments WS2, OR:WS2, PAA, and chlorine, caused a significant reduction in the initial bacteria counts, ranging from 0.67 to 1.82 log CFU/g. After 24 h, the OR2, WS2, PAA, and chlorine washing treatments effectively reduced bacteria from the leaf surface. After 7 days, the washing treatments OR2, WS2, OR:WS2, PAA, and chlorine effectively reduced initial microbial counts on butterhead lettuce. However, a slight increase in the bacterial population inoculated on butterhead lettuce was observed after 7 days when OR:WS1 (0.24 log CFU/g) and WS1 (0.14 log CFU/g) were used as sanitizer treatments (Appendix A).

For the storage duration of 0 h, OR:WS2 and chlorine treatments resulted in larger reductions (≥1.57 log CFU/g) in *E. coli* O157:H7 on inoculated romaine lettuce compared to the NR control. However, both treatments were statistically similar to the water control (*p* > 0.05). Contrarily, PAA and the other EO emulsions were less effective in reducing *E. coli* O157:H7, achieving reductions from 0.51 to 0.92 log CFU/g. The OR:WS2, PAA, and chlorine treatments resulted in significant reductions (*p* ≤ 0.05) in *E. coli* O157:H7 compared to the NR control and water for butterhead lettuce. In contrast, the OR1, OR2, WS1, WS2, and OR:WS1 treatments had no significant difference from the water and NR control (*p* > 0.05), with reductions ranging from 0.14 to 0.67 log CFU/g. Among the sanitizer treatments tested on crisphead lettuce, the OR2, OR:WS1, OR:WS2, PAA, and chlorine significantly reduced *E. coli* O157:H7 counts compared to the water and NR control, with reductions ranging from 2.51 to 2.77 log CFU/g. In contrast, the OR1, WS1, and WS2 resulted in similar results to water, with reductions ranging from 1.45 to 1.93 log CFU/g and no statistical differences (*p* > 0.05).

After 24 h of storage, all treatments significantly reduced *E. coli* O157:H7 inoculated on romaine and crisphead lettuce compared to the NR control (*p* ≤ 0.05). The OR2 caused a significantly higher reduction in bacteria inoculated on romaine lettuce compared to all other treatments and controls (NR and water), with a decrease of 1.97 log CFU/g. Chlorine had the second greatest reduction (1.61 log CFU/g), followed by OR1, OR:WS1, and WS1 with reductions of 1.41, 1.36, and 1.35 log CFU/g, respectively. The PAA treatment resulted in a reduction of only 0.62 log CFU/g, which was the lowest among all treatments. However, it caused a significantly greater decrease in bacteria on butterhead lettuce compared to water and NR control, with a reduction of 2.09 log CFU/g at 24 h. The OR1, WS1, and OR:WS1 had no significant difference from the NR control (*p* > 0.05) for butterhead lettuce. For crisphead lettuce, treatments that resulted in significantly higher reductions than all other treatments, water, and NR control were OR:WS1 and OR:WS2, with reductions of 3.52 and 3.41 log CFU/g, respectively.

After 7 days, chlorine resulted in significantly higher reductions in bacteria inoculated on romaine lettuce compared to all other treatments and controls (NR and water), with a reduction of 2.27 log CFU/g. Contrarily, the WS2 resulted in the lowest reduction, with only 0.61 log CFU/g reductions. For butterhead lettuce, the PAA resulted in significant reductions (*p* ≤ 0.05) in the bacteria population than all other tested treatments, with a decrease of 2.14 log CFU/g. Finally, for crisphead lettuce, all treatments were statistically similar to water (*p* > 0.05), except for OR1, which resulted in the lowest reduction in bacteria, with a decrease of 1.75 log CFU/g.

### 3.3. Cross-Contamination on Non-Inoculated Lettuce Leaves

The treatment effects on *E. coli* O157:H7 in NI lettuce samples were compared against sterile water, and the results are presented in Table 3. The log reductions in *E. coli* O157:H7 compared to the NR control obtained from surface NI lettuce are presented in Appendix A.

There was a significant interaction among sanitizer treatments, lettuce types, and storage times. 

The treatments of OR1, OR:WS2, and PAA effectively limited the *E. coli* O157:H7 cross-contamination from inoculated romaine leaves to washed NI romaine leaves over time (0 h, 24 h, and 7 days). After 7 days, the OR2 and WS2 were also effective in reducing bacteria cross-contamination in NI romaine lettuce, reaching levels below the limit of detection (LD). 

Treatments OR:WS2 and PAA also effectively reduced E. coli O157:H7 cross-contamination from inoculated butterhead leaves to NI butterhead leaves at 0 h. After 24 h and 7 days, both treatments (OR:WS2 and PAA) and OR2 successfully reduced the bacterial cross-contamination close to or below the LD on NI butterhead lettuce. 

At 0 h, treatments OR2, PAA, and chlorine effectively controlled *E. coli* O157:H7 cross-contamination from inoculated crisphead leaves to subsequently washed NI crisphead leaves. After 24 h and 7 days, treatments of OR2, PAA, chlorine, and OR:WS2 also effectively limited the bacteria cross-contamination (close to or below the LD) on NI crisphead lettuce. 

## 4. Discussion

The application of EO in food safety is limited, partly due to their low water solubility [14]. In the present study, Tween 80 was used as an emulsifier to increase the solubility of EOs and improve their antibacterial effects. Particularly, several studies have previously reported using Tween 80 as a surfactant in the formulation of EO emulsions [11,12,14]. The effectiveness of this approach can be associated with the MIC results of our study, in which OR, WS, and OR:WS had strong in vitro antimicrobial activity against *E. coli* O157:H7. In addition, the higher values of MBC compared to MIC agree with previous studies [22,41]. The winter savory EO utilized in this study predominantly comprised carvacrol (64.16%), gamma-terpinene (12.35%), and p-cymene (5.27%), while the oregano EO mainly contained carvacrol (71.17%), gamma-terpinene (5.20%), linalool (4.84%), and p-cymene (4.02%). However, it is noteworthy that the same plant species can yield EOs with distinct chemical compositions. The composition of EOs is influenced by various factors, including seasons, plant parts, storage, country of origin, extraction methods, and harvesting time [21,26,27,28,29]. The results of the current study show that OR had higher antimicrobial activity than WS. The higher concentration of carvacrol in OR compared to WS can, in part, explain its stronger antimicrobial activity. Carvacrol, the major component in both oregano and winter savory EOs, has demonstrated its efficacy against a wide range of pathogenic bacteria, including both Gram-positive and Gram-negative species [9,34,42,43,44,45,46,47,48,49,50,51], which further supports the findings of antimicrobial effect in this study [52]. Burt et al. [53] determined the MIC and MBC of carvacrol, thymol, gamma-terpinene, and p-cymene against *E. coli* O157:H7. The authors found that carvacrol and thymol exhibited inhibitory effects on the bacteria, while gamma-terpinene and p-cymene did not show satisfactory antimicrobial activity results. Despite the p-cymene’s lack of inhibitory activity when used alone, it was found to enhance the inhibitory effects of carvacrol when used in combination [47,54,55]. Moreover, p-cymene was observed to have a strong affinity for bacterial membranes, leading to membrane disruption and alterations in the membrane potential of intact cells [55]. Although p-cymene did not affect membrane permeability, it was suggested that this compound may decrease the enthalpy and melting temperature of the bacterial membrane, contributing to the antimicrobial activity of p-cymene-containing EOs [54]. 

Based on the results obtained from the MIC and MBC assays, the combination of OR and WS used in the present study improved the inhibitory activity of WS against *E. coli* O157:H7, suggesting an additive antimicrobial effect of EOs against *E. coli* O157:H7 in vitro. Similar studies have demonstrated the additive or synergistic antimicrobial effect of bioactive compounds from oregano and winter savory EOs [30,34,56,57,58,59]. Combining these components can significantly enhance their ability to inhibit microbial growth, with even the smallest addition of one component resulting in a significant increase in their action. Additionally, minor components of EOs might play a role in modulating their biological effects through synergistic effects [18]. The concentration of the oregano and winter savory EOs for the washing application was selected to be within the lowest MIC range (0.94 and 1.88 µL/mL). These results were comparable to those in the literature, with MIC values ranging from 0.72 to 2.48 mg/mL for oregano [18,60] and from 1.56 to 3.12 mg/mL for winter savory [18,41]. 

Although OR:WS2 and chlorine showed statistically similar results to those observed by the water control at 0 h, water was more effective in reducing attached *E. coli* O157:H7 from romaine leaves than all other sanitizer treatments. However, the reductions did not exceed 1.61 log CFU/g. In contrast, OR2, OR:WS1, OR:WS2, chlorine, and PAA performed significantly better than the water and NR control in reducing *E. coli* O157:H7 on crisphead, achieving more than 2.51 log CFU/g reductions. The antimicrobial effect of OR:WS2, chlorine, and PAA on butterhead was significantly better than the water and NR control. However, the reductions achieved by these treatments did not exceed 1.82 CFU/g. Chlorine and PAA findings in the present study agree with the results of previous studies, in which reductions were reported to be no more than 2.8 log CFU/g on contaminated produce in wash systems, including carrots [61], mung bean sprouts [6], lettuce [4], and peppers [7]. In a 2022 study, Cimowsky et al. [4] reported that 200 ppm chlorine and 80 ppm PAA achieved no more than 1.9, 2.4, and 2.8 log CFU/g reductions compared to an NR control on inoculated *E. coli* O157:H7 from romaine lettuce at 0 h, 24 h, and 7 days, respectively. In a study conducted by Dunn et al. [7], chlorine was not statistically different from the water control, resulting in less than 1 log CFU/g reduction on *Salmonella*-inoculated peppers (compared to an NR control).

During storage, there were significant differences in antimicrobial effectiveness among treatments. For romaine lettuce, the OR2 was the most effective sanitizer after 24 h of storage, resulting in a reduction of 1.97 log CFU/g, whereas PAA was the least effective, only achieving 0.62 log CFU/g reductions. After 7 days, chlorine became the most effective sanitizer for romaine lettuce, reaching a reduction of 2.27 log CFU/g. For crisphead lettuce, the OR:WS1 and OR:WS2 were significantly more effective than all other sanitizers, achieving reductions of 2.86 and 2.75 log CFU/g, respectively. However, after 7 days of storage, all treatments presented no significant difference from water, except for OR1, which resulted in a lower reduction in bacteria. These findings suggest that the efficacy of sanitizers can change over time, and the effectiveness of sanitizers can vary with the type of lettuce and the type of sanitizer used. Research conducted by Guerra-Rosas and colleagues [62] revealed that the antimicrobial activity of nanoemulsions containing oregano, thyme, and mandarin EOs against *E. coli* significantly decreased after 56 days of storage, which appears to be related to the depletion of volatile compounds. This same study demonstrated that freshly prepared oregano and thyme oil nanoemulsions resulted in a 2 log CFU reduction in *E. coli*; however, after 21 days of storage, the nanoemulsions only caused a 1 log CFU *E. coli* reduction, and even 7 days of storage resulted in carvacrol or thymol concentration reductions. The present study examined the antimicrobial effect from 0 h to 7 days; thus, the continued efficacy of the emulsified EOs (OR, WS, and OR:WS) compared to non-emulsified EOs from 0 h to 7 days suggests an advantage over single EOs. 

Regardless of concentration, the OR:WS1 decreased the bacteria population from inoculated crisphead lettuce at 0 h, 24 h, and 7 days. Future studies could examine lower concentrations of OR:WS1 to determine an optimal concentration for an OR:WS-based emulsion without losing its effectiveness for washing crisphead lettuce. Additionally, this lower concentration could be used as a washing treatment instead of OR:WS2 since neither has a statistical difference between the results for crisphead lettuce (*p* > 0.05).

Furthermore, the reduction in inoculated bacteria from the three lettuce types by the six EO emulsions was type- and dose-dependent. In general, after 7 days of storage, PAA and chlorine performed better than all EO treatments applied on inoculated romaine lettuce; OR2 and PAA performed better than chlorine used on inoculated butterhead lettuce; and OR2, OR:WS1, and OR:WS2 resulted in higher reductions in bacteria inoculated on crisphead lettuce than PAA and chlorine. The best results of emulsified EOs were obtained for those treatments prepared with an EO concentration of 0.94 µL/mL for romaine lettuce and 1.88 µL/mL for butterhead lettuce. Crisphead lettuce had a higher bacteria reduction when the OR concentration was 1.88 µL/mL and the WS and OR:WS concentrations were 0.94 µL/mL. Other studies also showed a concentration-dependent effect of EOs when they were applied to inhibit the growth of microorganisms on fresh produce [15,47,63,64,65]. 

Although the relationship between the surface characteristics of lettuce leaves and the washing effect of natural antimicrobial treatments is not yet completely understood, it has been demonstrated that surface roughness and hydrophobicity can significantly contribute to pathogen attachment and removal in fresh produce [66]. The rougher the surface of the lettuce, the more crevices and spaces for bacteria to attach and grow on the leaf. Compared to other lettuces, such as crisphead and butterhead, romaine lettuce has a relatively high surface roughness [67]; thus, romaine lettuce may have lower reductions in *E. coli* O157:H7 compared to smoother lettuce types such as butterhead and crisphead. In the present study, the PAA treatment had a higher decontamination effect on crisphead and butterhead than romaine lettuce during the 0 h and after 24 h. Chlorine had the highest antimicrobial activity in crisphead and similar results for butterhead and romaine lettuce at 0 h and after 24 h. At 7 days of storage, both PAA and chlorine significantly reduced *E. coli* O157:H7 in romaine lettuce. When EO emulsions were used as sanitizers, they were more effective in reducing *E. coli* O157:H7 populations on inoculated crisphead lettuce than romaine and butterhead. These findings indicate that the effectiveness of sanitizers can vary depending on the type of lettuce being treated. This is an important consideration for the produce industry, which relies on sanitizers to minimize the risk of foodborne illnesses. In addition, hydrophobic interactions between bacteria and lettuce surfaces are also a factor that facilitates bacteria attachment [68]. Therefore, the different inoculum preparation and inoculation methods, produce surface, produce storage time before treatment, method of sanitizer application, temperature difference between produce and sanitizer, recovery methods for surviving cells, and sample processing can all be used to explain the variable effectiveness of chlorine, PAA, and EO emulsions [69].

The present study also investigated the effect of sanitizers on preventing cross-contamination. The washing efficacy varied between the lettuce types, storage times, sanitizers, and concentrations. These results highlight the importance of selecting the appropriate sanitizer and concentration based on the specific conditions of the produce. PAA was the most effective treatment in limiting *E. coli* O157:H7 cross-contamination from inoculated lettuce leaves (romaine, crisphead, and butterhead) to subsequently washed NI lettuce leaves at all three storage times (0 h, 24 h, and 7 days). Chlorine was only effective in preventing cross-contamination on crisphead lettuce at all three storage times and on romaine at 0 h. The results suggest that PAA may be a better option than chlorine for controlling bacterial contamination during postharvest lettuce washing due to its superior efficacy in limiting cross-contamination. However, it is important to note that the effectiveness of sanitizers may vary depending on the lettuce type, storage time, and concentration. In some situations, chlorine may require a higher concentration or longer exposure time to be effective [4]. 

Among the EO emulsions tested, the results indicated that the effectiveness of EO emulsions in controlling cross-contamination in various lettuce types during different storage periods varied among the treatments. The OR:WS2 treatment effectively decreased bacterial populations over time on NI romaine and NI crisphead lettuce, with bacterial counts below the LD for NI butterhead lettuce. OR2 also effectively reduced *E. coli* O157:H7 on NI romaine and NI butterhead lettuce with bacterial counts below the LD on NI crisphead lettuce from 0 h to 7 days. WS1 treatment efficiently reduced *E. coli* O157:H7 on NI romaine, NI-butterhead, and NI crisphead lettuce over the three subsequent storage durations. Still, its effectiveness decreased after 24 h and 7 days, with no significant difference from water control during the washing of NI butterhead and crisphead lettuce. Particularly, the efficacy of some treatments, such as WS2 and OR:WS1, increased over time for some lettuce types, while the effectiveness of other treatments, such as OR1, decreased over time. For instance, the treatment WS2 performed significantly better at 24 h and 7 days than at 0 h for NI romaine and NI crisphead lettuce; however, it did not present significant differences from the water control at 24 h and 7 days for NI crisphead lettuce. OR:WS1 was significantly greater at 24 h and 7 days than at 0 h for NI romaine and NI butterhead lettuce, although it was not significantly different from the water control at 0 h, 24 h, and 7 days for NI butterhead lettuce. The treatment OR1 showed an increase in bacterial population over time when applied to NI butterhead lettuce, but it was not significantly different from water at 0 h, 24 h, and 7 days. For NI crisphead lettuce, OR1 was not significantly different from water at 24 h and 7 days. Thus, the efficacy of EO emulsions is not uniform and can vary depending on the EO emulsion, the type of lettuce, and the storage duration. Different lettuce types have different surface structures, which may affect the ability of the treatments to penetrate the surface and reach the bacterial cells [66,67]. Over time, the bacterial population may grow and adapt to the treatment conditions, which may affect their susceptibility to the treatments [70]. Additionally, the concentration of the EO emulsions in the washing solution could influence their efficacy over time, as the concentration of the active compounds could decrease over time due to degradation or other factors [71]. Therefore, these results demonstrate the variability in the effectiveness of different EO emulsions on different lettuce types and storage times. These findings highlight the importance of selecting the appropriate EO emulsion and concentration based on the specific conditions of the handled produce.

Despite having the lowest concentration, OR1 effectively controlled cross-contamination from inoculated romaine leaves to washed NI romaine leaves over time (0 h, 24 h, and 7 days). Therefore, lower concentrations could be analyzed to establish the minimum concentration for an OR-based emulsion while achieving similar efficacy to PAA for romaine lettuce. The WS2 was most effective after 7 days, suggesting that it may require a higher contact time to limit the cross-contamination of NI romaine leaves in washing produce systems. The same pattern was observed for OR2 applied on the butterhead surface and OR:WS2 applied on the crisphead surface. Both treatments were most effective after 24 h and 7 days, indicating that it may require a higher contact time with the lettuce surface to limit cross-contamination. There was no significant change in effectiveness over time for the treatments OR1 (NI romaine lettuce), WS2 and chlorine (NI butterhead lettuce), and OR:WS1 (NI crisphead lettuce) when examined over the three subsequent storage durations (0 h to 7 days). 

Overall, the results of this study provide important insights into using EO emulsions, PAA, and chlorine for microbial reduction in lettuce. It highlights the need to consider multiple factors, including the type of lettuce, storage duration, and EO emulsion concentration when selecting the most effective treatment. These findings can assist in the development of more targeted and effective interventions for reducing bacterial contamination in the postharvest washing of lettuce.

## 5. Conclusions

The development of natural sanitizers effective against foodborne bacteria is crucial for improving produce safety and providing sustainable options for producers. Emulsified EOs have shown promise as natural sanitizers for lettuce. In this study, oregano and winter savory EO emulsions were found to be effective in reducing bacterial populations on lettuce surfaces in a simulated washing system, with some formulations achieving similar or better results than traditional sanitizers such as chlorine and PAA. However, the effectiveness of the emulsified EOs varied depending on the lettuce type. These findings demonstrate that EO emulsions can be a viable alternative to traditional sanitizers for produce, but it is important to note that the concentration and formulation of the EO emulsions used in this study were optimized for the specific lettuce types and bacterial strains tested. Further research is needed to optimize the concentrations and formulations for different produce types and bacterial strains, as well as to evaluate the long-term effects of these sanitizers on produce quality and washing equipment. These results provide valuable insights into using EO emulsions to reduce bacterial populations on lettuce and can be used to inform best practices for handling and processing lettuce to ensure its safety and quality. Future research should focus on the economic viability and organoleptic effects of EO wash treatments and evaluate their long-term impacts on produce and equipment. Moreover, large-scale testing is required to assess the effectiveness of these natural sanitizers with higher throughputs.

## Figures and Tables

**Figure 1 foods-12-02571-f001:**
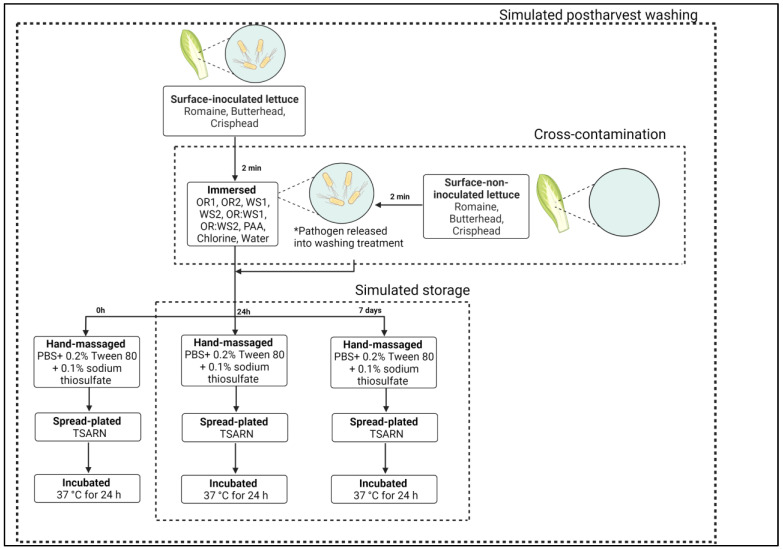
Schematic flow of simulated postharvest washing of lettuce in essential oil emulsion, 80 ppm PAA, 200 ppm chlorine, and sterile water at different storage times (0 h, 24 h, and 7 days). For each treatment, after washing the inoculated lettuce, one non-inoculated lettuce was immersed in the same washing solution to simulate cross-contamination from the initial contaminated sample. Abbreviations: OR: oregano; WS: winter savory; OR:WS: oregano and winter savory mixture; PAA: peroxyacetic acid. Essential oil emulsion concentrations: 1: 0.94 µL/mL; 2: 1.88 µL/mL. * The pathogen was released into the washing treatment after the inoculated lettuce leaves were washed.

**Table 1 foods-12-02571-t001:** Minimum inhibitory concentration and minimum bactericidal concentration of oil-in-water emulsions prepared with winter savory EO, oregano EO, and their combinations in different proportions against *E. coli* O157:H7.

Sample	MIC (µL/mL)	MBC (µL/mL)
WS	1.88	>30
OR	0.94	0.94
OR:WS (1:1)	0.94	7.50
OR:WS (3:1)	0.94	7.50
OR:WS (1:3)	0.94	3.75
OR:WS (2:3)	0.94	1.88
OR:WS (3:2)	0.94	7.50

Abbreviations: OR: oregano; WS: winter savory; OR:WS: oregano and winter savory mixture; MIC: minimum inhibitory concentration; MBC: minimum bactericidal concentration.

**Table 2 foods-12-02571-t002:** Log populations of *E. coli* O157:H7 recovered from surface-inoculated lettuce (romaine, butterhead, and crisphead) immediately after washing treatments (0 h) and after 24 h and 7 days of storage.

Treatments	Storage Times								
	Romaine (Log CFU/g)			Butterhead (Log CFU/g)			Crisphead (Log CFU/g)		
No Treatment	0 h	24 h	7 Days	0 h	24 h	7 Days	0 h	24 h	7 Days
NR	4.56 ± 0.28	4.61 ± 0.03	4.45 ± 0.09	5.71 ± 0.27	5.65 ± 0.10	5.64 ± 0.10	5.80 ± 0.22	5.69 ± 0.06	5.44 ± 0.02
Water	3.04 ± 0.01	3.93 ± 0.07	4.64 ± 0.22	5.20 ± 0.06	5.48 ± 0.42	5.79 ± 0.14	4.24 ± 0.21	4.07 ± 0.11	2.94 ± 0.50
Sanitizers									
OR1	3.64 ± 0.03 ^b,A^	3.22 ± 0.15 ^c,d,B^	3.09 ± 0.17 ^b,c,B^	5.56 ± 0.21 ^a,A^	5.42 ± 0.37 ^a,A^	5.47 ± 0.26 ^a,A^	4.35 ± 0.18 ^a,A^	4.05 ± 0.21 ^a,A^	3.69 ± 0.01 ^a,B^
OR2	4.05 ± 0.07 ^a,A^	2.64 ± 0.12 ^e,C^	3.36 ± 0.09 ^b,B^	5.07 ± 0.22 ^a,A^	4.85 ± 0.22 ^b,A^	3.83 ± 0.13 ^c,d,B^	3.03 ± 0.06 ^c,A^	2.66 ± 0.05 ^d,B^	2.46 ± 0.15 ^d,B^
WS1	3.67 ± 0.03 ^a,b,A^	3.26 ± 0.08 ^c,d,B^	2.96 ± 0.08 ^c,C^	5.57 ± 0.18 ^a,A^	5.76 ± 0.30 ^a,A^	5.78 ± 0.16 ^a,A^	4.17 ± 0.07 ^a,b,A^	3.51 ± 0.06 ^b,B^	2.76 ± 0.07 ^b,c,C^
WS2	3.80 ± 0.11 ^a,b,A^	3.80 ± 0.34 ^a,b,A^	3.84 ± 0.33 ^a,A^	5.04 ± 0.17 ^a,A^	5.22 ± 0.20 ^a,b,A^	4.24 ± 0.29 ^b,c,B^	3.87 ± 0.20 ^b,A^	3.80 ± 0.05 ^a,b,A^	2.84 ± 0.05 ^b,c,B^
OR:WS1	3.88 ± 0.05 ^a,b,A^	3.25 ± 0.00 ^c,d,B^	3.07 ± 0.04 ^b,c,B^	5.25 ± 0.41 ^a,B^	5.38 ± 0.05 ^a,B^	5.88 ± 0.23 ^a,A^	3.29 ± 0.04 ^c,A^	2.17 ± 0.23 ^e,C^	2.58 ± 0.09 ^c,d,B^
OR:WS2	2.99 ± 0.13 ^c,B^	3.56 ± 0.15 ^b,c,A^	3.38 ± 0.16 ^b,A^	4.17 ± 0.05 ^b,B^	4.80 ± 0.05 ^b,A^	4.33 ± 0.29 ^b,B^	3.15 ± 0.05 ^c,A^	2.28 ± 0.12 ^e,C^	2.69 ± 0.05 ^b,c,d,B^
80 ppm PAA	3.70 ± 0.14 ^a,b,A^	3.99 ± 0.24 ^a,A^	2.57 ± 0.13 ^c,B^	3.89 ± 0.18 ^b,A^	3.56 ± 0.29 ^d,B^	3.50 ± 0.06 ^d,B^	3.18 ± 0.02 ^c,A^	3.14 ± 0.07 ^c,A^	2.97 ± 0.16 ^b,A^
200 ppm Chlorine	2.95 ± 0.06 ^c,A^	3.00 ± 0.12 ^d,A^	2.18 ± 0.06 ^d,B^	4.09 ± 0.05 ^b,A^	4.14 ± 0.38 ^c,A^	4.27 ± 0.17 ^b,A^	3.10 ± 0.18 ^c,A^	3.01 ± 0.01 ^c,A^	2.90 ± 0.12 ^b,A^
Contrasts									
Sanitizers vs. Water									
OR1 X Water	***	***	***	ns	ns	ns	ns	ns	***
OR2 X Water	***	***	***	ns	*	***	***	***	ns
WS1 X Water	***	***	***	ns	ns	ns	ns	***	ns
WS2 X Water	***	ns	***	ns	**	***	ns	ns	ns
OR:WS1 X Water	***	***	***	ns	ns	ns	***	***	ns
OR:WS2 X Water	ns	ns	***	***	ns	***	***	***	ns
80 ppm PAA X Water	***	ns	***	***	***	***	***	***	ns
200 ppm Chlorine X Water	ns	***	***	***	***	***	***	***	ns
Sanitizers vs. NR									
OR1 X NR	***	***	***	ns	ns	ns	***	***	***
OR2 X NR	***	***	***	ns	**	***	***	***	***
WS1 X NR	***	***	***	ns	ns	ns	***	***	***
WS2 X NR	***	***	***	*	**	***	***	***	***
OR:WS1 X NR	***	***	***	ns	ns	ns	***	***	***
OR:WS2 X NR	***	***	***	***	**	***	***	***	***
80 ppm PAA X NR	***	***	***	***	***	***	***	***	***
200 ppm Chlorine X NR	***	***	***	***	***	***	***	***	***
No treatment									
Water X NR	***	***	ns	ns	ns	ns	***	***	***

Values followed by similar lowercase letters indicate no significant differences (*p* > 0.05) among treatments (rows) within storage times and lettuce types (columns). Values followed by similar uppercase letters indicate no significant differences (*p* > 0.05) among storage times (columns) within treatments (rows) and lettuce types (columns). Abbreviations: NR: no-rinse OR: oregano; WS: winter savory; OR:WS: oregano and winter savory mixture; PAA: peroxyacetic acid. EO emulsion concentrations: 1: 0.94 µL/mL; 2: 1.88 µL/mL. ns: nonsignificant. *, **, ***: significant at *p* ≤ 0.05, 0.01, 0.001, respectively.

**Table 3 foods-12-02571-t003:** Log populations of *E. coli* O157:H7 recovered from surface non-inoculated lettuce (romaine, butterhead, and crisphead) immediately after treatments (0 h) and throughout 24 h and 7 days of storage.

Treatments	Storage Times								
	Romaine (Log CFU/g)			Butterhead (Log CFU/g)			Crisphead (Log CFU/g)		
No Treatment	0 h	24 h	7 Days	0 h	24 h	7 Days	0 h	24 h	7 Days
Water	2.88 ± 0.16	1.75 ± 0.09	1.80 ± 0.02	1.98 ± 0.50	2.33 ± 0.03	2.81 ± 0.06	3.40 ± 0.30	2.17 ± 0.13	1.66 ± 0.07
Sanitizers									
OR1	0.24 ± 0.05 ^c,d,A^	0.26 ± 0.02 ^d,e,A^	0.18 ± 0.15 ^c,d,A^	1.69 ± 0.04 ^c,d,B^	2.02 ± 0.08 ^b,A,B^	2.36 ± 0.25 ^a,b,A^	1.85 ± 0.24 ^b,A^	1.45 ± 0.03 ^b,B^	1.46 ± 0.05 ^b,B^
OR2	0.74 ± 0.26 ^c,A^	0.97 ± 0.28 ^c,A^	<LD ^d,B^	0.91 ± 0.08 ^d,A^	<LD ^d,B^	0.13 ± 0.10 ^d,B^	<LD ^e,A^	<LD ^d,A^	<LD ^d,A^
WS1	2.15 ± 0.53 ^a,A^	1.95 ± 0.03 ^a,A^	0.42 ± 0.19 ^b,c,B^	4.12 ± 0.27 ^a,A^	2.91 ± 0.27 ^a,B^	2.81 ± 0.48 ^a,B^	2.64 ± 0.17 ^a,A^	1.64 ± 0.00 ^b,B^	1.29 ± 0.07 ^b,C^
WS2	1.42 ± 0.20 ^b,A^	0.45 ± 0.22 ^d,B^	<LD ^d,C^	1.54 ± 0.43 ^c,d,A^	1.24 ± 0.34 ^c,A^	1.18 ± 0.17 ^c,A^	2.56 ± 0.07 ^a,A^	2.23 ± 0.15 ^a,A,B^	1.93 ± 0.19 ^a,B^
OR:WS1	1.83 ± 0.07 ^a,b,A^	1.15 ± 0.15 ^b,c,B^	0.56 ± 0.15 ^b,C^	2.91 ± 0.21 ^b,A^	2.59 ± 0.25 ^a,A,B^	2.12 ± 0.12 ^a,b,B^	1.07 ± 0.13 ^c,A^	0.76 ± 0.30 ^c,A^	0.60 ± 0.18 ^c,A^
OR:WS2	0.57 ± 0.13 ^c,d,A^	<LD ^e,B^	<LD ^d,B^	<LD ^e,A^	<LD ^d,A^	<LD ^d,A^	0.45 ± 0.01 ^d,A^	0.14 ± 0.09 ^d,B^	<LD ^d,C^
80 ppm PAA	<LD ^d,A^	<LD ^e,A^	<LD ^d,A^	<LD ^e,A^	<LD ^d,A^	<LD ^d,A^	<LD ^e,A^	<LD ^d,A^	<LD ^d,A^
200 ppm Chlorine	1.55 ± 0.14 ^a,b,A^	1.46 ± 0.06 ^b,A^	1.09 ± 0.05 ^a,B^	1.88 ± 0.65 ^c,A^	2.42 ± 0.12 ^a,b,A^	1.60 ± 0.06 ^b,c,A^	<LD ^e,A^	<LD ^d,A^	<LD ^d,A^
Contrasts									
Sanitizers vs. Water									
OR1 X Water	***	***	***	ns	ns	ns	***	ns	ns
OR2 X Water	***	*	***	**	***	***	***	***	***
WS1 X Water	ns	ns	***	**	ns	ns	*	ns	ns
WS2 X Water	**	***	***	ns	***	***	*	ns	ns
OR:WS1 XWater	ns	ns	***	ns	ns	ns	***	***	***
OR:WS2 X Water	***	***	***	***	***	***	***	***	***
80 ppm PAA X Water	***	***	***	***	***	***	***	***	***
200 ppm Chlorine X Water	*	ns	ns	ns	ns	*	***	***	***

Values followed by similar lowercase letters indicate no significant differences (*p* > 0.05) among treatments (rows) within storage times and lettuce types (columns). Values followed by similar uppercase letters indicate no significant differences (*p* > 0.05) among storage times (columns) within treatments (rows) and lettuce types (columns). Abbreviations: OR: oregano; WS: winter savory; OR:WS: oregano and winter savory mixture; PAA: peroxyacetic acid; LD: limit of detection: −1 log CFU/g. EO emulsion concentrations: 1: 0.94 µL/mL; 2: 1.88 µL/mL. ns: nonsignificant.; *, **, ***: significant at *p* ≤ 0.05, 0.01, 0.001, respectively.

## Data Availability

Data is contained within the article or Appendix A.

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
