# Peer review of "Use of Essential Oil Emulsions to Control Escherichia coli O157:H7 in the Postharvest Washing of Lettuce"

_foods, 2023, doi:10.3390/foods12132571_

Round 1

Reviewer 1 Report

Comments to authors

Foods-2383034

In the research, Use of Essential Oil Emulsions as an Alternative Management to Control Escherichia coli O157:H7 in the Postharvest Washing of Lettuces, scientifically interesting data is available. The results and the discussion parts are well written, authors have elaborated on all the points. I recommend this article for publication after improving some comments and questions listed below. I hope my comments will be of help to improve the manuscript.

1.     The language of the manuscript needs a drastic revision and the authors should revise the language of the whole manuscript very carefully.

2.     The abstract needs to be improved and well structured. There are some grammatical mistakes in the abstract and in the manuscript that could be hard to understand by readers. For example Line, 22-23.

3.     The introduction part needs drastic improvements. There is very small data on the importance of EO emulsions and the need for this study should also be elaborated in this part.

4.     A workflow diagram of this study must be added for easy understanding by readers.

5.     Line 93-94 “A 10-µL loopful was…..” use correct preposition.

6.     Section 5, conclusions, re-write it, it must contain cocluding remarks but not the summary.

1The language of the manuscript needs a drastic revision and the authors should revise the language of the whole manuscript very carefully.

Author Response

May 11, 2023

Dear Reviewer 1

Foods

Subject: Response letter

Manuscript ID: foods-2383034

Title: Use of Essential Oil Emulsions as an Alternative Management to Control

Escherichia coli O157:H7 in the Postharvest Washing of Lettuces

            Thank you for your valuable feedback on our manuscript. We have carefully reviewed your comments and suggestions and have made the necessary revisions to improve the abstract and manuscript's grammar and structure. We have also thoroughly proofread the manuscript and made sure that all grammatical errors have been corrected to ensure that the manuscript is well-written and easy to understand for the readers. Once again, we appreciate your time and effort in reviewing our manuscript, and we hope you will find the revised version satisfactory for publication in Foods.

            As a general explanation, any changes in the revised version are highlighted in yellow. Please see the attachment. 

Reviewer 1’s comments:

In the research, Use of Essential Oil Emulsions as an Alternative Management to Control Escherichia coli O157:H7 in the Postharvest Washing of Lettuces, scientifically interesting data is available. The results and the discussion parts are well written, authors have elaborated on all the points. I recommend this article for publication after improving some comments and questions listed below. I hope my comments will be of help to improve the manuscript.

  1. The language of the manuscript needs a drastic revision and the authors should revise the language of the whole manuscript very carefully.

Response: The manuscript has been thoroughly proofread, and all grammatical errors have been corrected to ensure that the manuscript is well-written and easy to understand for the readers. The manuscript was finally revised by an external English native speaker and by all co-authors, who have an extensive record o peer-review publications in international scientific journals and federally funded grants.

  1. The abstract needs to be improved and well structured. There are some grammatical mistakes in the abstract and in the manuscript that could be hard to understand by readers. For example Line, 22-23.

Response: The necessary changes have been made in the Abstract to improve the abstract’s grammar and structure per the reviewer’s suggestions.

  1. The introduction part needs drastic improvements. There is very small data on the importance of EO emulsions and the need for this study should also be elaborated in this part.

Response: The Introduction section has been revised to provide a more detailed explanation of the importance of EO emulsion and the need for this study. Additional information was included in the introduction section to approach the reviewer’s concern.

  1. A workflow diagram of this study must be added for easy understanding by readers.

Response: A schematic flow diagram has been added to the Materials and Methods section to provide more detailed information regarding the study design.

  1. Line 93-94 “A 10-µL loopful was…..” use correct preposition.

Response: The necessary correction has been made: A 10-µL loopful was transferred to fresh TSBRN.

  1. Section 5, conclusions, re-write it, it must contain cocluding remarks but not the summary.

Response: The Conclusion section has been revised and restructured to focus on the key findings and their implications for the use of essential oil emulsions as natural sanitizers for lettuce.

Reviewer 2 Report

The subject of this paper, according to authors, was ‘to evaluate the antimicrobial efficacy of emulsions of winter savory EO and oregano EO in comparison with no-rinse treatment, water, chlorine, and PAA in a simulated washing system against E. coli O157:H7 using three lettuce types (romaine, butterhead, and crisphead).

The objective in this research paper is really interesting and deals with the potential use of alternative sanitizers for postharvest management of lettuce.

The Introduction part is well written and gives a brief overview of the problem addressed, as well the advantages of alternative sanitizers. The structure is well laid out and the abstract reflects the key elements of the article content. Additionally, I believe that the manuscript has the novelty and the originality required for the specific scientific Journal.

Nevertheless, there are some issues, that need to be addressed by authors. I would suggest to change the word ‘management; within the title and replace it by another, more suitable word (eg method/approach,..). Is the application of such agents (chlorine/PAA, EO etc) mostly aiming at a surface decontamination? I believe that this issue should be better clarified throughout the text.

Another point of interest is related to legislative matters; is the use of EO (and in particular the hydrophobic part) allowed for use in such foods, such as lettuce? It would be helpful to provide information on which are the sanitizers permitted to be used at this point. Are there specific limits of their application?

 Other points to be addressed:

Line 140, pg3: on what basis were those two concentrations (0.94 and 1.88) selected?

Lines 144-145, pg 13: Can you explain this contradictory trend over time for the efficacy of the different treatments?

Referring to the microbial reduction obtained (for example in line 183 pg 14, and in numerous parts of this submission), a reduction of about 2.5 logs is deemed adequate for sanitizing purposes?  Another point of crucial importance is the sensory attributes gained by applying those EO. Essential oils, especially coming out of oregano, are well known to have strong flavors/taste/aromas. Changes induced in EO treated samples are acceptable, from a sensory viewpoint?

Author Response

May 11, 2023

Dear Reviewer 2

Foods

Subject: Response letter

Manuscript ID: foods-2383034

Title: Use of Essential Oil Emulsions as an Alternative Management to Control

Escherichia coli O157:H7 in the Postharvest Washing of Lettuces

            Thank you for your valuable feedback on our manuscript. We have carefully reviewed your comments and suggestions and have made the necessary revisions to improve the abstract and manuscript's grammar and structure. We have also thoroughly proofread the manuscript and made sure that all grammatical errors have been corrected to ensure that the manuscript is well-written and easy to understand for the readers. Once again, we appreciate your time and effort in reviewing our manuscript, and we hope that you will find the revised version to be satisfactory for publication in Foods.

            As a general explanation, any changes done in the revised version are highlighted in yellow. Please see the attachment.

Reviewer 2’s comments:

The subject of this paper, according to authors, was ‘to evaluate the antimicrobial efficacy of emulsions of winter savory EO and oregano EO in comparison with no-rinse treatment, water, chlorine, and PAA in a simulated washing system against E. coli O157:H7 using three lettuce types (romaine, butterhead, and crisphead).

The objective in this research paper is really interesting and deals with the potential use of alternative sanitizers for postharvest management of lettuce.

The Introduction part is well written and gives a brief overview of the problem addressed, as well the advantages of alternative sanitizers. The structure is well laid out and the abstract reflects the key elements of the article content. Additionally, I believe that the manuscript has the novelty and the originality required for the specific scientific Journal.

Nevertheless, there are some issues, that need to be addressed by authors.

  1. I would suggest to change the word ‘management; within the title and replace it by another, more suitable word (eg method/approach,..).

Response: The title has been revised and modified according to reviewer’s suggestion. The new title is “Use of Essential Oil Emulsions to Control Escherichia coli O157:H7 in the Postharvest Washing of Lettuces”.

  1. Is the application of such agents (chlorine/PAA, EO etc) mostly aiming at a surface decontamination? I believe that this issue should be better clarified throughout the text.

Response: According to the U.S. Federal Regulation for fresh produce, the application of antimicrobial substances (e.g., chlorine, PAA and EOs) is mostly aimed at limiting cross-contamination during the postharvest washing step. Although there might be some considerable microbial population reduction from the produce surface, it is not considered a safety measure to prevent illnesses.

https://www.federalregister.gov/documents/2015/11/27/2015-28159/standards-for-the-growing-harvesting-packing-and-holding-of-produce-for-human-consumption

  1. Another point of interest is related to legislative matters; is the use of EO (and in particular the hydrophobic part) allowed for use in such foods, such as lettuce? It would be helpful to provide information on which are the sanitizers permitted to be used at this point. Are there specific limits of their application?

Response: The FDA (U.S. Food and Drug Administration) does not provide specific recommendations for the upper limit of human consumption of essential oils. In general, the FDA regulates essential oils as food additives or flavorings, and they are generally recognized as safe (GRAS) when used in accordance with their intended use. However, the FDA does not regulate essential oils as dietary supplements or for therapeutic use, and it is important to note that not all essential oils are safe for internal use. In this study, both oregano and winter savory essential oils are regulated as GRAS by the FDA. A paragraph was included in the Introduction section to highlight this point.

  1. Line 140, pg3: on what basis were those two concentrations (0.94 and 1.88) selected?

Response: Emulsion concentrations were selected based on the Minimum inhibitory concentration (MIC) results. This information was provided in the 2.5 Preparation of washing treatments and results discussed in the manuscript.

  1. Lines 144-145, pg 13: Can you explain this contradictory trend over time for the efficacy of the different treatments?

Response: The discussion was revised, and additional information was added in the Discussion section. The contradictory trend observed in the study's results can be explained by the fact that the effectiveness of the EO emulsions varied based on the type of lettuce and the storage time. Different lettuce types have different surface structures, which may affect the ability of the treatments to penetrate the surface and reach the bacterial cells. Over time, the bacterial population may grow and adapt to the treatment conditions, which may affect their susceptibility to the treatments. Additionally, the concentration of the EO emulsions in the washing solution could have an effect on their efficacy over time, as the concentration of the active compounds could decrease over time due to degradation or other factors.

  1. Referring to the microbial reduction obtained (for example in line 183 pg 14, and in numerous parts of this submission), a reduction of about 2.5 logs is deemed adequate for sanitizing purposes?  

Response: The acceptable level of reduction may vary depending on the specific industry and product being treated. A reduction of 2.5 to 3 logs (99.7 to 99.9%) in microorganisms can be considered adequate for sanitizing purposes, depending on the initial level of contamination and the intended use of the lettuce. In some cases, a higher reduction may be necessary, such as when the lettuce will be consumed by vulnerable populations such as the elderly, pregnant women, or immunocompromised individuals. However, as highlighted in question 2, the main purpose of adding antimicrobial agents in postharvest washing is not to reduce microbial load but to prevent cross-contamination.

  1. Another point of crucial importance is the sensory attributes gained by applying those EO. Essential oils, especially coming out of oregano, are well known to have strong flavors/taste/aromas. Changes induced in EO treated samples are acceptable, from a sensory viewpoint?

Response: It depends on the specific essential oil used and the concentration applied. Essential oils such as oregano and winter savory have strong flavors and aromas that can potentially affect the sensory attributes of the treated samples. However, suppose the concentration of the essential oil is carefully controlled. In that case, especially when used in lower amounts, it is possible to minimize any negative impact on the sensory quality of the lettuce. Further research will be done to optimize the concentrations and formulations for different produce types and bacterial strains, as well as to evaluate the long-term effects of these sanitizers on produce quality and washing equipment.

Reviewer 3 Report

Dear Editor and authors, 

A- Major comments 

1- What the bioactive compounds in used essential oils? Such manuscripts cannot be accepted without knowing the active compounds in the inhibitory substances. The authors should have used several techniques to detect these important compounds such as GC-mass, HPLC, and Spam-GCMS.

B- Minor comments

1-The introduction of the manuscript needs to some modifications, such as add new paragraph or section about effect of essential oils on foodborne pathogens. Please read these paper 

-de Almeida, J. M., Crippa, B. L., de Souza, V. V. M. A., Alonso, V. P. P., Júnior, E. D. M. S., Picone, C. S. F., ... & Silva, N. C. C. (2023). Antimicrobial action of Oregano, Thyme, Clove, Cinnamon and Black pepper essential oils free and encapsulated against foodborne pathogens. Food Control, 144, 109356.

-Al-Fekaiki, D. F., Niamah, A. K., & Al-Sahlany, S. T. G. (2017). Extraction and identification of essential oil from Cinnamomum zeylanicum barks and study the antibacterial activity. Journal of Microbiology, Biotechnology and Food Sciences, 7(3), 312-316.

-Mantzourani, I., Daoutidou, M., Dasenaki, M., Nikolaou, A., Alexopoulos, A., Terpou, A., ... & Plessas, S. (2022). Plant Extract and Essential Oil Application against Food-Borne Pathogens in Raw Pork Meat. Foods, 11(6), 861.

2-Please add new reference of Preparation of EO emulsion method (line 70-76).

3-Many method need to add references such as Simulated postharvest treatment, Simulated storage, and Cross-contamination in non-inoculated lettuce leaves.

4-What is the active compound in the essential oils used in the research? The presence of these active compounds was not discussed. What is the concentration of these compounds?

5-The extracted essential oils from the same plant vary in the quality and concentration of active substances depending on the plant's age and the extraction method, among other factors. How can the authors recommend the use of these oils in preserving lettuce leaves?

6-The conclusions contain many results that were previously mentioned in the Results section. The results should be removed and clear conclusions for the manuscript should be rewritten.

The language in the manuscript is good, the grammar is good, and the manuscript does not need a linguistic evaluation.

Author Response

May 11, 2023

Dear Reviewer 3

Foods

Subject: Response letter

Manuscript ID: foods-2383034

Title: Use of Essential Oil Emulsions as an Alternative Management to Control

Escherichia coli O157:H7 in the Postharvest Washing of Lettuces

            Thank you for your valuable feedback on our manuscript. We have carefully reviewed your comments and suggestions and have made the necessary revisions to improve the abstract and manuscript's grammar and structure. We have also thoroughly proofread the manuscript and made sure that all grammatical errors have been corrected to ensure that the manuscript is well-written and easy to understand for the readers. Once again, we appreciate your time and effort in reviewing our manuscript, and we hope that you will find the revised version to be satisfactory for publication in Foods.

            As a general explanation, any changes done in the revised version are highlighted in yellow. Please see the attachment.

Reviewer 3’s comments:

A- Major comments 

  1. What the bioactive compounds in used essential oils? Such manuscripts cannot be accepted without knowing the active compounds in the inhibitory substances. The authors should have used several techniques to detect these important compounds such as GC-mass, HPLC, and Spam-GCMS.

Response: Although the objective of this research was not to evaluate the chemical composition of the essential oils yet to test the synergistic effect of the oil blends, the principal investigator obtained a GC/MS report from each supplier to provide information on the oil quality and chemical composition and ensure the concentration range of the primary compounds was in accordance with other publications. A statement has been included in the materials and methods section for clarification purposes. Other relevant publications in the same area have also reported the same approach:

https://www.nature.com/articles/s41598-017-08673-9

https://www.sciencedirect.com/science/article/pii/S0362028X22097861?via%3Dihub

B- Minor comments

  1. The introduction of the manuscript needs to some modifications, such as add new paragraph or section about effect of essential oils on foodborne pathogens. Please read these paper 

-de Almeida, J. M., Crippa, B. L., de Souza, V. V. M. A., Alonso, V. P. P., Júnior, E. D. M. S., Picone, C. S. F., ... & Silva, N. C. C. (2023). Antimicrobial action of Oregano, Thyme, Clove, Cinnamon and Black pepper essential oils free and encapsulated against foodborne pathogens. Food Control, 144, 109356.

-Al-Fekaiki, D. F., Niamah, A. K., & Al-Sahlany, S. T. G. (2017). Extraction and identification of essential oil from Cinnamomum zeylanicum barks and study the antibacterial activity. Journal of Microbiology, Biotechnology and Food Sciences, 7(3), 312-316.

-Mantzourani, I., Daoutidou, M., Dasenaki, M., Nikolaou, A., Alexopoulos, A., Terpou, A., ... & Plessas, S. (2022). Plant Extract and Essential Oil Application against Food-Borne Pathogens in Raw Pork Meat. Foods, 11(6), 861.

Response: The Introduction section has been revised, and modifications have been made to provide a more detailed explanation of the effect of essential oils on foodborne pathogens. Two new paragraphs about this topic were added to the Introduction.

  1. Please add new reference of Preparation of EO emulsion method (line 70-76).

Response: The Materials and Methods section has been revised, and references have been included for the preparation of the EO emulsions method used in our study, which was scientifically validated.

  1. Many method need to add references such as Simulated postharvest treatment, Simulated storage, and Cross-contamination in non-inoculated lettuce leaves.

Response: The Materials and Methods section has been revised, and references have been included for the Simulated postharvest treatment, Simulated storage, and Cross-contamination in non-inoculated lettuce leaves, which have been scientifically validated.

  1. What is the active compound in the essential oils used in the research? The presence of these active compounds was not discussed. What is the concentration of these compounds?

Response: The active component in the winter savory EO and oregano EO and their respective concentration range have been added in the Introduction section. The major component of oregano EO is carvacrol, with concentrations ranging from 61 to 85%. Similarly, winter savory EO contains carvacrol at similar ranges (55 to 89%) as the main component.

  1. The extracted essential oils from the same plant vary in the quality and concentration of active substances depending on the plant's age and the extraction method, among other factors. How can the authors recommend the use of these oils in preserving lettuce leaves?

Response: The main purpose of this study is to evaluate the efficacy of emulsions containing oregano and winter savory EOs at different concentrations to be used in the postharvest washing. It is true that the quality and concentration of active substances in EOs vary depending on several factors. However, it is also true that the range of the main components in EOs is relatively consistent for a particular plant species. Therefore, it is possible to recommend using certain EOs, such as oregano and winter savory, for postharvest washing lettuce leaves, as their antibacterial properties have been well established through previous research.

  1. The conclusions contain many results that were previously mentioned in the Results section. The results should be removed and clear conclusions for the manuscript should be rewritten.

Response: The Conclusion section has been revised and restructured to focus on the key findings and their implications for using essential oil emulsions as natural sanitizers for lettuce.

Round 2

Reviewer 3 Report

Dear Editors , 

The authors did not provide clear answers to the main question of the study

What are the active compounds found in used essential oils?

It is known that the composition of essential oils varies according to seasons, method of manufacture, methods of extraction, country of origin and many other factors. Here we do not know the compounds present in the essential oils and the concentration of these active substances in the oil, so we cannot recommend the use of the oil in the lettuce washing water. Because the compounds in the oil are not stable and their concentrations are different. The study cannot be accepted on speculative results.

Manuscript language is good.
